# IR-Band Conversion of Target and Background Using Surface Temperature Estimation and Error Compensation for Military IR Sensor Simulation

**DOI:** 10.3390/s19112455

**Published:** 2019-05-29

**Authors:** Taewuk Bae, Youngchoon Kim, Sangho Ahn

**Affiliations:** 1Daegu-Gyeongbuk Research Center, Electronics and Telecommunications Research Institute, Daegu 42994, Korea; twbae@etri.re.kr; 2Department of Information and Communication Security, U1 University, Asan 31415, Korea; 3Department of Electronic Telecommunications, Mechanical & Automotive Engineering, Inje University, Kimhae 50834, Korea; elecash@inje.ac.kr

**Keywords:** infrared image, wavelength, object surface, temperature estimation, radiance

## Abstract

Military infrared (IR) imaging systems utilize one or more IR wavelength-bands, among short wavelength IR (SWIR), middle wavelength IR (MWIR), and long wavelength IR (LWIR) band. The IR image wavelength-band conversion which transforms one arbitrary IR wavelength-band image to another IR wavelength-band image is needed for IR signature modeling and image synthesis in the IR systems. However, the IR wavelength-band conversion is very challenging because absorptivity and transmittance of objects and background (atmosphere) are different according to the IR wavelength band and because radiation and reflectance characteristics of the SWIR are very different from the LWIR and MWIR. Therefore, the IR wavelength-band conversion in this paper applies to only IR targets and monotonous backgrounds at a long distance for military purposes. This paper proposes an IR wavelength-band conversion method which transforms one arbitrary IR wavelength-band image to another IR wavelength-band image by using the surface temperature estimation of an object and the error attenuation method for the estimated temperature. The surface temperature of the object is estimated by an approximated Planck’s radiation equation and the error of estimated temperature is corrected by using the slope information of exact radiance along with the approximated one. The corrected surface temperature is used for generating another IR wavelength-band image. The verification of the proposed method is demonstrated through the simulations using actual IR images obtained by thermal equipment.

## 1. Introduction

Infrared (IR) imaging is widely used in military, medical, and industrial field since it is able to create a visual with otherwise non-visible wavelength band to the human eye [1,2]. Recently, IR imaging systems, such as IR search and track (IRST), IR countermeasures (IRCM), and directional IR countermeasure (DIRCM) etc, have been used for developing and evaluating military IR imaging systems. The military applications of IR imaging system are increasing due to the fact that it provides quick verification and cost reduction in concept understanding, principle establishment, and performance assessment for the IR system function. In order to analyze these existing IR systems or future IR systems, IR targets and backgrounds should be modeled theoretically as IR signatures with brightness or intensity values corresponding to temperature for those IR images. Many methods for modeling IR signatures have been developed, based on mechanical theories such as the radiation principle, atmospheric transfer characteristics, and heat transfer theory [3,4,5,6,7]. In addition, many commercial software tools for modeling IR signatures were also developed, for example, RadThermIR [8] and MuSES [9] for predicting IR signatures and Vega Prime Sensor [10], OKTAL-SE [11], and vieWTerra [12] for virtual-reality simulation under an IR environment. However, the IR signature modeling software has shown many limitations regarding security and is also costly. Generally, the IR signature modeling tool and imaging system use the characteristics of short wavelength IR (SWIR), middle wavelength IR (MWIR), and long wavelength IR (LWIR) to improve the military and industrial utility. For example, the DIRCM system for missile defense requires simulations from viewpoint of an airplane as well as an IR-guided missile [13,14,15,16]. These simulations require IR images of three IR wavelength bands (SWIR, MWIR, and LWIR) to enhance the defense capability against the missile by using the synthesized image of the target and the background of various IR wavelength bands [17]. However, obtaining three IR wavelength-band images comes with substantial challenge. For example, thermal imagers for each wavelength band should be equipped and same object or scene should be taken at the same time. Therefore, to overcome the difficulty of acquiring IR images in each wavelength band, an IR wavelength-band transformation method that transforms an IR image in arbitrary IR wavelength band to IR images in another wavelength band is required.

In the case of the IR-guided missile [18,19,20], a reticle seeker using a single detector uses SWIR-band and MWIR-band, on the other hand, an imaging seeker mainly uses LWIR-band [6,21,22]. Therefore, IR images from the three IR wavelength bands are required for these simulations. In addition, the DIRCM system for missile defense requires simulations for IR-guided missiles as well as an IR image-tracker that tracks missiles on airplanes [13,23,24]. Additionally, for this purpose, target detection methods for detecting small targets or moving objects in IR images are used, because missiles are usually included in backgrounds of IR images. Since the IR-guided missiles are common in modern warfare, various spatial [25,26,27,28] and temporal [1,29,30] target-detection methods have been proposed for detecting missiles or small targets. The IR image tracker uses IR images of LWIR-band or MWIR-band. Therefore, IR images of all three IR wavelength bands are required for simulating the entire DIRCM system. However, it is difficult to acquire IR images of three IR wavelength bands at the same time because the thermal equipment of each IR wavelength band is always needed.

This paper proposes a method for IR wavelength-band conversion which transforms arbitrary IR wavelength-band image to other IR wavelength-band image based on the surface temperature estimation process of an object and the error attenuation method for the estimated temperature. The surface temperature of an object is estimated by approximating the Planck’s radiation intensity formula including an integral operator. Then, the estimated temperature is compensated for by the error attenuation technique by using the slope information of exact radiance and approximated one. The compensated temperature is used to convert the IR image of the original wavelength band into an IR image of the desired wavelength band. The proposed temperature estimation process and the error compensation technique are analyzed quantitatively through simulation, and verification is confirmed through the IR wavelength-band transformation experiment for various IR background images obtained from an IR camera. In detail, the modeled MWIR images made by an IR signature modeling tool and actual MWIR images are compared with the converted MWIR images by the proposed method from their original LWIR images.

The proposed IR band conversion method may simplify the construction of IR system that consist of various IR band cameras [31,32] such as a forward-looking infrared (FLIR) system for obtaining LWIR and MWIR image used for target tracking and detection. This method can be applied to the aforementioned small target detection field [25,26,27,28,29,30] by using temperature, radiation, and gray-level information obtained from the plurality of IR band images converted from arbitrary IR band images and can be utilized for accurate classification of military targets such as tanks, planes, and warships [33,34,35]. In addition, in IR and the visible image synthesis field [36] for object detection, the detection performance can be further improved by combining converted IR band images.

## 2. Materials and Methods

### 2.1. Relationship of Temperature and Radiance

As a black body is an ideal IR radiator, by means of Planck’s law, L(λ,T), the spectral radiance of the black body for specific wavelength λ and temperature T is given by as follows:(1)$$L(\lambda ,T)=\frac{C_{1}}{\lambda^{5}\left[\exp(C_{2}/\lambda T)-1\right]}\left[W/cm^{2}\mu msr\right],$$
where $C_{1}=1.191\times 10^{4}\left[W\mu m^{4}/cm^{2}sr\right]$ and $C_{2}=1.428\times 10^{4}\left[\mu mK\right]$ are radiance constants [37,38,39]. The radiance intensity of an IR wavelength band width ($\lambda_{1}\sim \lambda_{2}$) for an object is given
(2)$$L(T)=\int_{\lambda_{1}}^{\lambda_{2}}\frac{\varepsilon (\lambda )C_{1}}{\lambda^{5}\left[\exp(C_{2}/\lambda T)-1\right]}d\lambda \left[W/cm^{2}sr\right],$$
where ε(λ) is the spectral emissivity of the object, ε(λ)=1 for the black body and ε(λ)<1 for the gray body. Since the relationship between the temperature and the radiance of Equation (2) includes an integral operator, it is very difficult to inversely estimate the temperature from the radiance. Therefore, the radiance calculation formula of Equation (2) is approximated as follows in this paper:(3)$$L(T)≐\frac{\varepsilon C_{1}\Delta \lambda }{\lambda_{c}^{5}\left[\exp(C_{2}/\lambda_{c}T)-1\right]},$$
where ε is the emissivity and we assume that it is a constant, namely, is the gray body in the wavelength band. $\lambda_{c}(=(\lambda_{1}+\lambda_{2})/2)$ and $\Delta \lambda (=\lambda_{2}-\lambda_{1})$ represent the center wavelength in the wavelength band and the wavelength-band width. Because the approximated radiance calculation formula of Equation (3) is unnecessary to integrate, the calculation is simple. So, the temperature T can be easily restored through the inverse transformation from the radiance L(T). As the radiant energy in the arbitrary IR wavelength band is absorbed by oxygen, carbon dioxide, ozone, and vapor among an atmosphere, it is not detected in specific wavelength band. According to these atmosphere transmission feature, a wavelength range generally used in an IR system is 1.9~2.9 μm for the SWIR, 3~5 μm for MWIR, and 8~12 μm for LWIR, respectively.

Figure 1 shows the spectral radiance according to the temperature for a black body in the wavelength, obtained from Planck’s law of Equation (1). And the radiance intensity according to temperature obtained through Equation (2) for the three IR wavelength bands (SWIR, MWIR, and LWIR) are shown in Figure 2. In case area of a target seen in an IR detector is $A_{p}$ and the solid angle of the detector at the target is Ω, the output voltage $V_{\det}$ from the detector is given by:(4)$$V_{\det}=A_{p}\Omega L(T)\int_{\lambda_{1}}^{\lambda_{2}}\tau_{amb}(\lambda )\tau_{opt}(\lambda )S(\lambda )d\lambda [V],$$
where $\tau_{amb}(\lambda )$ and $\tau_{opt}(\lambda )$ are the spectral transmittance of the atmosphere and the detector optics respectively. S(λ) is the spectral responsivity of the detector with the unit [V/W]. If it assumes that $\tau_{amb}(\lambda )$, $\tau_{opt}(\lambda )$, and S(λ) are a time-invariant system, the integral values of the wavelength band have a constant value. $A_{p}$ and Ω also are a constant. This means that the output voltage of the detector is proportional to the radiance L(T) according to the temperature of the target. As shown in Equation (2), the radiance L(T) is the function of the spectral emissivity ε(λ), however, as it becomes constant ε for the black body or the gray body. Therefore, the radiance L(T) becomes the function of the only temperature T. For a two-dimensional IR imaging detector, the gray-level of the IR image is also proportional to the output voltage of the detector. Therefore, we can assume that L(T), the radiance of specific wavelength band for an object temperature T, is proportional to the gray-level of the IR image. When performing the band transformations with the aforementioned assumptions in terms of spectral transmittance and emissivity, we can deduce the following conclusions: 

Generally, the IR wavelength-band conversion is an extremely difficult and challenging area of research, because materials have different emissivity, reflectivity, and transmissivity that varies with the IR wavelength band. This variance is strongly affected by water vapor (H_2_O) and carbon dioxide (CO_2_) among atmospheric gases because these components have a specific absorption/emission spectrum according to the wavelength band. Although the current work includes conversions to the SWIR band as well as the LWIR and MWIR, the absorption and transmission of an object in the SWIR-band are affected by the active reflectivity of photons, unlike the middle wave using passive thermal radiation. Therefore, in this paper, the IR wavelength band conversion between the LWIR and MWIR are mainly performed and analyzed for metallic targets commonly used in the military (relatively well-known helicopter, airplane, tank, and ship etc.) and analyzed for monotonous backgrounds (sky, ground, and sea background images of less than 10 Km from an observer) for the IRCM or IRST. To reduce the aforementioned complexity of the IR wavelength band, the emissivity and atmospheric transmissivity of the object, which are external parameters of the above mentioned parameters, are simplified based on the following facts. Now, we will discuss the transmissivity of the atmosphere and the emissivity of the object in the wavelength band.

First, it is necessary to set the transmittance of H_2_O and CO_2_ since it has the greatest effect on the atmospheric transmittance. Generally, the IR bands are affected by the concentration of H_2_O and CO_2_. From troposphere (altitude less than 11 km) to ground, the concentration of two gases in a function of height in the atmosphere is, respectively, about 0.0003 VMR (volume mixing ratio, ppm unit) for CO_2_ and 0~0.008 VMR for H_2_O. Because most of IR images used in the IRCM simulator were taken under 10 km from the ground, we assume the concentration is constant. In addition, the transmittance of major atmospheric gases including H_2_O and CO_2_ in the MWIR and LWIR band is nearly closed to 1, whereas it is irregular in the SWIR-band. Thus, assuming that transmissivity of the atmosphere ($\tau_{amb}$) is uniform, we can speculate that conversion between the LWIR and MWIR band is acceptable, however, conversion between another band and the SWIR band may be a bit inaccurate.

Second, emissivity of a target (object) or background should be set to vary with wavelength. Many materials are not gray body with the emissivity independent of wavelength. In the IRCM or IRST system, a criterion for dividing a target and background image is either metal (including ferrous metals and non-ferrous metal) or non-metal. Generally, the emissivity of the metal decreases exponentially as the wavelength increases. More in detail, the emissivity of the metal abruptly decreases from 75% to 25% for 0 to 2.5 μm (in the SWIR-band), then decreases very slowly from about 20% to 10% for 2.5 to 14 μm (from the MWIR to LWIR band). The emissivity of the non-metal is about 30% in the SWIR-band, about 50%~85% on average in the MWIR band, and about 90% in the LWIR-band. In here, we can know that the emissivity of the metal (or non-metal) is high in the MWIR- and LWIR-band, but is much smaller in the SWIR-band. Thus, assuming that an object or background is the black body or gray body, we can speculate that the conversion between the LWIR and MWIR is understandable, but that the conversion between the SWIR-band and another band may be a bit inaccurate.

Finally, one should pay attention to the conversion accuracy between wavelengths. Because the LWIR and MWIR detect the thermal radiation emitted from the object itself, while the NIR and SWIR use active near infrared (NIR)/SWIR reflectivity of the object. The conversion between other IR bands and the SWIR-band may contain some errors than the conversion between the LWIR-band to MWIR-band. Therefore, we focus more on the conversion between the LWIR- and MWIR-band in this paper.

### 2.2. Wavelength-Band Conversion based on Temperature Estimation and its Error Compensation

Figure 3 shows the flow chart of the proposed wavelength-band conversion method. The proposed method first assigns min, max temperature and radiance to the min, max gray-level pixels in an arbitrary input IR wavelength band image, and then estimates the radiance and temperature for gray-levels of all remaining pixels. Using the estimated temperature, the radiance of the desired IR band is obtained and then converted to a gray level.

#### 2.2.1. Wavelength-Band Conversion Using Temperature Estimation

We propose an IR wavelength-band conversion method which transforms arbitrary IR wavelength-band image to another IR wavelength-band one, based on the temperature estimation of an object. Figure 4 shows the transformation relation between radiance and gray-level of the IR image according to the temperature of an object. In case the object is the gray-body (i.e., the emissivity is constant ε), the shape of the radiance curve is similar with that of the black body (ε=1.0); however, the size decreases in proportion to emissivity. In particular, the figure shows an example that a pixel of maximum gray-level $G_{\max}$ represents an object having maximum temperature $T_{\max}$ and emissivity of 0.8 and a pixel of minimum temperature $T_{\min}$ represents an object having minimum gray-level $G_{\min}$ and emissivity of 0.6 in the acquired IR image. First, by using Equation (3), the relation formula for estimating the temperature T from the radiance L is derived as follows:(5)$$\begin{array}{l} L(T)≐\frac{\varepsilon C_{1}\Delta \lambda }{\lambda_{c}^{5}\left[\exp(C_{2}/\lambda_{c}T)-1\right]}\\ \to \exp(C_{2}/\lambda_{c}T)-1≐\frac{\varepsilon C_{1}\Delta \lambda }{L\lambda_{c}^{5}}\\ \to \ln\left[\exp(C_{2}/\lambda_{c}T)\right]≐\ln\left[\frac{\varepsilon C_{1}\Delta \lambda }{L\lambda_{c}^{5}}+1\right]\\ \to C_{2}/\lambda_{c}T≐\ln\left[\frac{\varepsilon C_{1}\Delta \lambda }{L\lambda_{c}^{5}}+1\right]\\ \to T≐\frac{C_{2}}{\lambda_{c}\ln\left(\frac{\varepsilon C_{1}\Delta \lambda }{L\lambda_{c}^{5}}+1\right)} \end{array}$$

Using the relationship between radiance and gray-level according to temperature in Figure 3, the gray-level transfer function G(L) from the radiance intensity can be represented by:(6)$$G(L)=a[L-L(T_{\min})]+G_{\min},$$
where a is the slope of conversion function that is obtained by:(7)$$a=\frac{G_{\max}-G_{\min}}{L(T_{\max})-L(T_{\min})},$$
where the radiance for $T_{\max}$ corresponding to the pixel value $G_{\max}$ is as the following:(8)$$L(T_{\max})≐\varepsilon \frac{C_{1}}{\lambda_{c}^{5}\left[\exp(C_{2}/\lambda_{c}T_{\max})-1\right]}\Delta \lambda ,$$
The radiance for $T_{\min}$ corresponding to the pixel value $G_{\min}$ is as the following:(9)$$L(T_{\min})≐\varepsilon \frac{C_{1}}{\lambda_{c}^{5}\left[\exp(C_{2}/\lambda_{c}T_{\min})-1\right]}\Delta \lambda ,$$
The radiance L(G) from the gray-level G using Equation (6) can be inversely calculated by:(10)$$L(G)=\frac{G-G_{\min}}{a}+L(T_{\min}),$$

It is assumed that an IR image for arbitrary wavelength band is given and the temperature and emissivity of an object corresponding to the gray-level of at least two pixels in the image are known. The larger the difference of the temperature and the gray-level between the two pixels, the better the transformation result. The procedure of wavelength-band conversion using temperature estimation is as follows:
Step I: Calculate *max and min radiance* using gray-level-to-temperature assignment and temperature-to-radiance relation.In an original (source) IR image of arbitrary IR wavelength band to convert, when assuming that maximum, minimum gray-level $G_{\max}$, $G_{\min}$ are assigned to maximum, minimum temperature $T_{\max}$, $T_{\min}$ and these radiance intensities $L(T_{\max})$ and $L(T_{\min})$ can be obtained using the temperature-radiance function of Equation (3).Step II: Obtain *temperature for all the pixel* using gray-level-to-radiance and radiance-to-temperature relation.The radiance intensity L(G) for the gray-level G for all the pixel in the original IR image is obtained using Equation (10). Temperature T corresponding to the obtained radiance intensity for all the pixel is calculated through Equation (5).Step III: Obtain *radiance of desired IR wavelength band* for all the pixel using temperature-to-radiance relation.The radiance intensity of the desired (destination) IR wavelength band for all the pixel of the original IR image is calculated by Equation (3) using the temperature T obtained in the step II and the emissivity ε.Step IV: Generate *desired IR wavelength-band image* using radiance-to-gray-level relation.Finally, the desired IR wavelength-band image is generated by the radiance-to-gray-level transfer function G(L) using the radiance intensity of the desired IR wavelength band obtained in the step III for all the pixel of the original IR image. The gray-level transfer function is given by:(11)$$G(L)=g\frac{255(L-L_{\min})}{L_{\max}-L_{\min}}+b,0\leq G(L)\leq 255,$$
where g(0<g≤1) is the contrast controlling constant and b is the brightness controlling constant. For g=1 and b=0, the gray-level of the desired IR wavelength-band image is distributed at 0~255.

#### 2.2.2. Error Analysis and Compensation for Estimated Temperature

Since the proposed IR wavelength-band conversion method using the surface temperature of an object uses Equation (3) approximating the radiance formula of Equation (2) including an integral operator, the computation become simple, however, causes an error of the estimated temperature. Figure 5 shows the exact and approximated radiance value according to the temperature obtained by Equation (2) and Equation (3) for three IR wavelength bands in 200 K~600 K temperature range. Figure 6 represents the errors of the estimated temperature relative to the original temperature for three IR wavelength bands. The temperature estimation error represents the difference between the original temperature and the estimated one. As shown in the figure, the radiance intensity and the estimated temperature error have respective different characteristics per the wavelength band.

Figure 7 represents exact and approximated radiance according to arbitrary temperature. The exactly calculated radiance $L^{*}(T)$ is obtained by Equation (2) including an integral operator, on the other hand, the approximated radiance L(T) is calculated by Equation (3) approximated. $\hat{T}$ represents the estimated temperature obtained from the radiance $L(\hat{T})$ using Equation (5) and $\tilde{T}$ represents the compensated temperature. The iteration formula for calculating the exact temperature from the approximately estimated temperature is as follows:(12)$$\begin{array}{ll} {\tilde{T}}_{j+1} & ={\tilde{T}}_{j}-\mu \cdot e({\tilde{T}}_{j})\cdot {\left[\frac{dL^{*}({\tilde{T}}_{j})}{d{\tilde{T}}_{j}}\right]}^{-1}\\ & ={\tilde{T}}_{j}-\mu \cdot \{L^{*}({\tilde{T}}_{j})-L({\tilde{T}}_{j})\}\cdot {\left[\frac{dL^{*}({\tilde{T}}_{j})}{d{\tilde{T}}_{j}}\right]}^{-1},\;j=0,1,2,\ldots \end{array}$$
where ${\tilde{T}}_{0}=\hat{T}$ and μ is the convergence factor. The *j*-th compensated temperature ${\tilde{T}}_{j}$ is updated using the estimated temperature error $e({\tilde{T}}_{j})$ and the tangent slope of the optimal radiance, and then ${\tilde{T}}_{j}$ is repeatedly calculated until it converges without any difference with ${\tilde{T}}_{j+1}$, the *j*+1-th compensated temperature. This method can obtain the optimal temperature, however, it has the disadvantage of high computational complexity because repeated computation for obtaining the optimal temperature is required. Therefore, the compensated temperature $\tilde{T}$ obtained using only the first order tangent slope of $L^{*}(\hat{T})$ for $\hat{T}$ is as follows:(13)$$\tilde{T}=\hat{T}-\frac{2\Delta T\{L^{*}(\hat{T})\}}{L^{*}(\hat{T}+\Delta T)-L^{*}(\hat{T}-\Delta T)},$$
where $L^{*}(\hat{T})$ is the exact radiance for the estimated temperature and ΔT represents the micro-temperature displacement value to obtain the tangential slope at the *A* point in the figure.

### 2.3. Dataset Used for Experiment

The IR images used in the simulation were obtained with TAU640, a LWIR camera of the FLIR corporation. First, photographed CAR (270 K~322 K) and CUP (271 K~340 K) images containing high temperature objects are used to prove the compensation effect of the approximately estimated temperature. Actual photograph of eight sky background LWIR images (SKY1~SKY8), five sea (or river) background LWIR images (SEA1~SEA5), and five ground background LWIR images (GND1~GND5) with various temperature and gray-level distributions converted to MWIR and SWIR bands and analyzed. LWIR images (SHIP, TANK, and MOUNTAIN) modeled by IR signature and background simulation tools and actual images taken with LWIR (BUILDING, AIRPORT, GROUND, and TANK2) and are converted into MWIR images by the proposed method. Then, the converted MWIR images are compared with the originally modeled MWIR images and the actual photographed MWIR images through gray-level difference and peak signal to noise ratio (PSNR).

## 3. Results and Discussion

The feasibility of the temperature estimation and its error compensation method for the proposed IR wavelength-band conversion is demonstrated through the simulation. Figure 8 shows the approximately estimated temperature $\hat{T}$, the compensated temperature $\tilde{T}$, and their temperature error with the exact temperature $T^{*}$ (given by Equation (2)) for the three IR wavelength bands. The temperature error represents the difference between the approximately estimated temperature (or the compensated temperature) and exact temperature. The compensated temperature $\tilde{T}$ was obtained through Equation (13). The approximately estimated temperature error ${\hat{e}}_{T}$ is given by
(14)$${\hat{e}}_{T}=\hat{T}-T^{*},$$
The compensated temperature error ${\tilde{e}}_{T}$ is given by:(15)$${\tilde{e}}_{T}=\tilde{T}-T^{*},$$

In the case of the SWIR-band, as shown in Figure 8a, the approximate estimated temperature is higher than compensated temperature for 200 K to 600 K. The error was larger at relatively low temperature, but decreased as temperature increases. The error of the compensated temperature was greatly reduced compared to the error of the approximately estimated temperature, however, at 200 K, the error of 9 K occurred even if the approximately estimated temperature was compensated for. In the case of MWIR-band, as shown in Figure 8b, the sign of the approximately estimated temperature error ${\hat{e}}_{T}$ was changed at about 420 K, and the error after the temperature compensation was greatly reduced. At 200 K, the error of 3 K occurred even after the temperature compensation. In the case of the LWIR-band, as shown in Figure 8c, the sign of the approximately estimated temperature error was changed at about 410 K, and the compensated temperature error greatly decreased at all temperatures.

In order to verify the feasibility of the proposed IR wavelength-band conversion method, the IR wavelength-band conversion was simulated for IR images as shown in Figure 9. The IR images used in the simulation were obtained with TAU640, a LWIR camera of FLIR corp., and the acquired IR images (CAR and CUP) are shown in Figure 9a. In the CAR image, the engine part of the vehicle has the brightest gray-level (pixel value of 255). On the other hand, the sky background region has the lowest gray-level (pixel value of 0). These temperatures were 322 K and 270 K respectively, and were measured using an IR thermometer. The CUP image was obtained by filling the paper-cups with water at different temperatures and filling the cups on the lower right corner with ice. The temperatures of the pixels having the maximum brightness (pixel value of 255) and the minimum brightness (pixel value of 0) are 340 K and 271 K, respectively. Figure 9b,c shows the pseudo color images of the approximately estimated temperatures and its compensated temperature respectively for the CAR and CUP images. In the case of the LWIR-band, as shown in Figure 8c, the approximately estimated temperature is lower than the compensated temperature from 200 K to 400 K, so approximately estimated temperature of Figure 9b has more blue color in the pseudo color, compared to the compensated temperature of Figure 9c. Figure 9d,e represents images obtained by converting the LWIR images (CAR and CUP image) into MWIR and SWIR images according to the compensated temperatures, respectively.

The IR wavelength-band conversion is generally converted from LWIR image to MWIR and SWIR image. LWIR image is easy to acquire, and the gray-level distribution according to temperature for the LWIR image is close to relatively linear compared to other IR wavelength bands. On the other hand, SWIR image has little change in image brightness because radiance at low temperature region is very small. That is, even if an IR image of SWIR-band is converted into IR images of high wavelength bands such as the MWIR- and LWIR-band, it may not be an effective conversion. Therefore, it may not be appropriate to use the SWIR image as an input source for IR wavelength-band conversion. In estimating and compensating a temperature of an object or a background from a MWIR image, it is reasonable to use it at about 260 K or more because it has an error only at a very low temperature of 200K as shown in Figure 8b.

Figure 10 and Figure 11 show the result of converting the test LWIR images composed of three groups, i.e., sky background images (SKY1~SKY8), sea background images (SEA1~SEA5), and ground background images (GND1~ GND5), into MWIR and SWIR images. As shown in the figures, the IR wavelength-band conversion was performed with the marked max, min temperature and gray-level. As shown in the MWIR and SWIR images converted from the original LWIR images by the proposed method, the brightness of the converted MWIR and SWIR images was compressed because the conversion function from LWIR- to MWIR-, SWIR-band (namely, from high to low wavelength bands) has an exponential function form. All the background images used in the experiment are in the temperature range of 200 K to 400 K, so, we can also confirm this, and the pseudo color images of the approximately estimated temperature is more blue than that of the compensated temperature images.

In the sky background images of Figure 10, SKY1 has the narrow temperature range and pixel distribution with low temperature, while SKY2~SKY 8 have a relatively wide temperature range (temperature difference of more than 100 degrees) and pixel distribution (almost 0 to 255). In the converted MWIR and SWIR images of the original LWIR sky images, the regions with high temperature, marked with the red arrow in the figures, are highlighted by the increased radiance, for examples, the bright front cloud region in SKY1, the village region at the bottom right in SKY2, the upper partial cloud region in SKY3, the bottom front cloud region in SKY4, the roadside structure region and the low mountain region in SKY5, the front leaf region in SKY6, the tree branch region and the structure region of the road in SKY7, and the distant mountain region and the structure region of the road in SKY8. On the other hand, the regions with lower temperature, marked with the blue arrow in the figures, are darkened due to the decrease in the radiance, for examples, the high sky regions in almost all images (also the bottom cloud region in SKY3). We can see that since the temperature of the ground or tree is higher than the sky background, the pixel value of the ground or tree region is expanded due to the increase of the radiance in the regions, whereas the pixel value of the sky region is compressed because of the radiance decrease in the region. We can also see that the detail of the image increases due to the expansion and compression effect of the radiance as the image of the high wavelength band is converted into the image of the low wavelength band. It can also be seen that the approximately estimated temperature error is larger than the compensated temperature error and thus appears to be bluer as shown in the temperature pseudo-image of the sky region.

In the sea background images of Figure 11, while SEA1 has the narrow temperature range and pixel distribution with many dark pixels due to low temperature, SEA2~SEA5 has the wide temperature range and pixel distribution. In the converted MWIR and SWIR images for the original LWIR images, the high temperature regions marked with the red arrow (the low right ground region in SEA1, the bottom ground region in SEA2 and SEA3, and the hill region with structures and trees in SEA4 and SEA5) are highlighted by the increase in the radiance. On the other hand, the low temperature regions (the sky or the sea or the river region in all sea background images) marked with the blue arrow are darkened due to the decrease in the radiance. We can know that radiation brightness of the sky or the sea (or the river) region decreases because the temperature of the sea (or the river) region is lower than the ground region, when the sea and the ground region are in one image. Similar to the experimental results for the sky background images, the contrast ratio of the ground or the structure regions with higher temperature is improved due to the radiance increase effect, whereas the contrast ratio of the sea or the river regions with lower temperature is decreased due to the radiance decrease effect. Also, in the river or sea region, the error of the compensated temperature is less than the error of the approximately estimated temperature, so it can be seen that it is less blue. 

In the ground background images of Figure 11, because GND1 and GND5 contain the land (or soil) region with very high temperature, it is much brighter than other images. GND2, GND3, and GND4 are dark on average due to the regions with the low temperature. In the converted MWIR and SWIR images, the high temperature regions (the land region in all images) marked with the red arrow are highlighted, while the low temperature regions (the sky and the tree region in all images) marked with the blue arrow are reduced in the brightness. We can know that since the temperature is usually high in the order of sky, tree (or forest), ground, and structure, the radiance-increased effect occurs in that order at the IR wavelength-band conversion. In addition, it is possible to confirm that the temperature of the entire image is slightly increased as the estimated temperature error is compensated.

Figure 12 shows actual LWIR, MWIR images (BUILDING, AIRPORT, GROUND, TANK2) and the modeled LWIR, MWIR images (SHIP, TANK produced by OKTAL-SE of OKTAL corp. [11] and MOUNTAIN produced by vieWTerra of VWORLD corp. [12]) and the converted MWIR images by the proposed method from the original LWIR images through the temperature to radiance curve and the gray-level transfer function. We can see that the specific objects and the background regions are well transformed, for example: 1) the body region of the ship and the tank composed of metal, the front sea region, and the hill region in the converted SHIP, TANK, and TANK2; 2) the distant mountain and the front mountain slope region in MOUNTAIN; and 3) the structure or the object region in the converted BUILDING, AIRPORT, GROUND, and TANK2. On the other hand, the sky regions in each the converted MWIR images is not well transformed compared to the original MWIR image, for example, the sky region of BUILDING and AIRPORT. This is because the absorption and emission rate of atmospheric gases, especially water vapor, ozone, and carbon dioxide differ according to the IR wavelength band, especially between LWIR- and MWIR-band. Also, in the sky region of the LWIR images, it is implied that the saturated (namely, covered) region by the low transmissivity is difficult to convert (i.e., restore) to the sky region of the original MWIR.

Figure 13 shows the pixel values of the red, green, and blue lines on the original MWIR image and the converted MWIR image for SHIP, TANK, MOUNTAIN, BUILDING, AIRPORT, GROUND, TANK2 image of Figure 12. In the converted SHIP image of Figure 12a, the pixels corresponding to the ship body region (green line and dotted line, metal part) and the sea region (blue line) are transformed to be similar to the pixels of the regions of the original MWIR image, but the pixels of the sky region (red line and dotted line) are different from the original MWIR pixels. Similarly, in the converted TANK image of Figure 12b, the pixels corresponding to the warehouse region (green line and dotted line) and the tank body region (blue line and dotted line, metal part) are well converted, but the sky pixels (red line and dotted line) are not. This is because it is difficult to reconstruct the sky region because it depends on the absorption, scattering, and transmissivity of the atmospheric gases according to the IR wavelength bands as mentioned above. On the other hand, in the case of MOUNTAIN image of Figure 12c with the monotonous sky region, it can be seen that the pixel values corresponding to the distant mountain region (red line and dotted line) and the mountainside region (green line and dotted line) of the mountain are well transformed. Likewise, in the case of the TANK2 image of Figure 12g, the pixel values corresponding to the forest region (red line and dotted line) and the tank body region (blue line and dotted line) are transformed to be similar to the original MWIR pixel values. The aforementioned conversion problem for the sky region occurs similarly in the comparison between the actual MWIR image and the converted MWIR image for each test images. In each converted of Figure 12d–f, the converted structure region (green line and dotted line) and the ground region (blue line and dotted line) in BUILDING of Figure 12d, AIRPORT of Figure 12e, and GROUND of Figure 12f are very similar to the original pixels, but the sky pixel (red line and dotted line) is not restored properly. This is more noticeable in Figure 12d than in Figure 12e,f. This means that Figure 12d is more difficult to recover because the concentration, as well as absorption and transmissivity of atmospheric gases is more complex than Figure 12e,f. Table 1 shows the PSNR results for the red, green, and blue lines in Figure 13. We can confirm that the IR wavelength-band conversion of the object or target region (almost metal in IR sensor simulations), the monotonous region (almost H_2_O), and the ground region (almost soil) are more accurate compared to the sky region.

## 4. Conclusions

This paper proposes an IR wavelength-band conversion method which transforms arbitrary IR wavelength-band image to another IR wavelength-band image based on the surface temperature estimation of an object and the error attenuation technique of the estimated temperature. This IR band conversion method can solve the difficulty of building an expensive and complex IR camera system required to obtain IR images of various IR bands in the FLIR or IRST system. Since this method can create IR images of other bands only with IR images of a specific band obtained from a single IR camera, it can be applied to industrial and military fields to analyze IR radiation information emitted by various objects in multiple bands. In particular, the radiation and gray-level information of objects in various IR bands become an important key to detecting and tracking military threats. Moreover, IR images of various IR bands obtained by this method can also be utilized in the recently studied IR and visible image-fusion field for high image reliability and understandability.

## Figures and Tables

**Figure 1 sensors-19-02455-f001:**
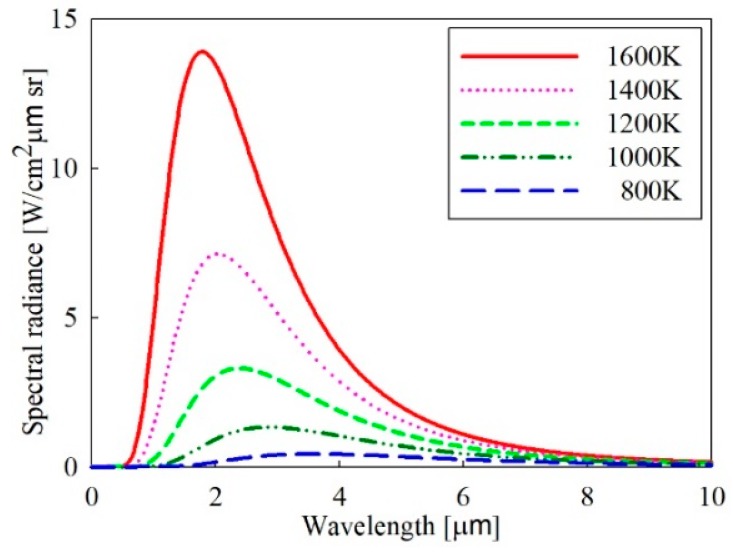
Spectral radiance according to temperature for a black body.

**Figure 2 sensors-19-02455-f002:**
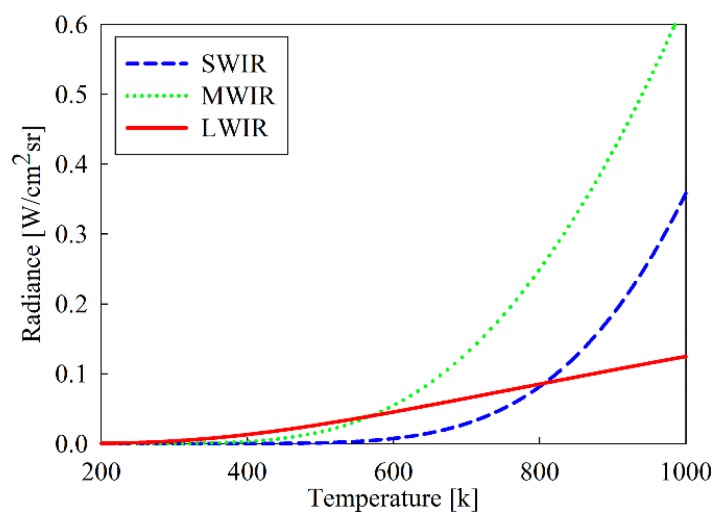
Radiances of three infrared (IR) wavelength bands according to temperature.

**Figure 3 sensors-19-02455-f003:**
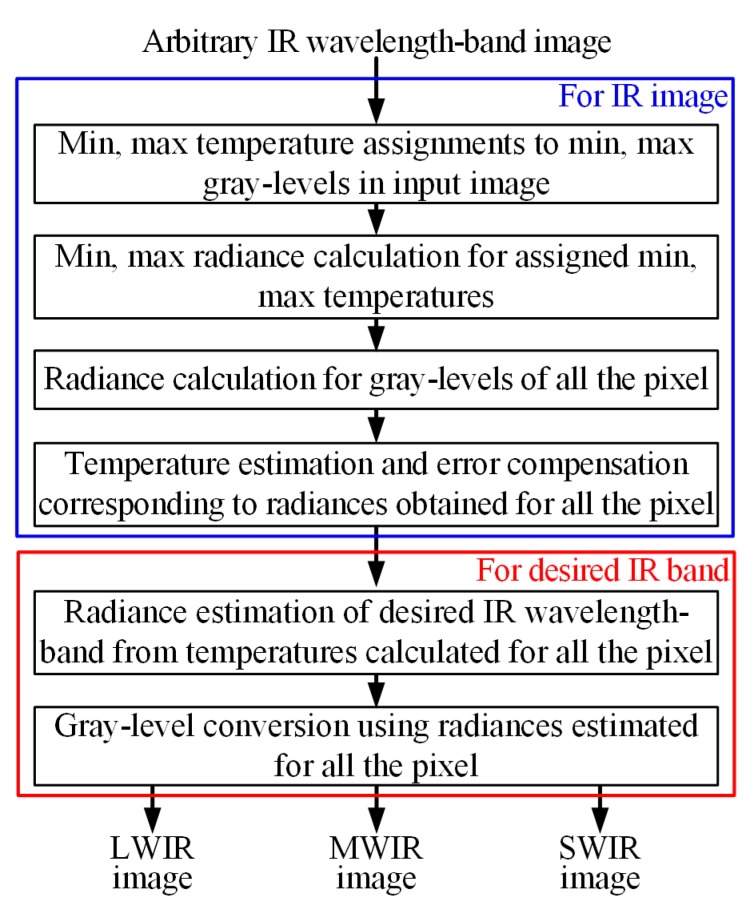
Flow chart of proposed wavelength-band conversion method.

**Figure 4 sensors-19-02455-f004:**
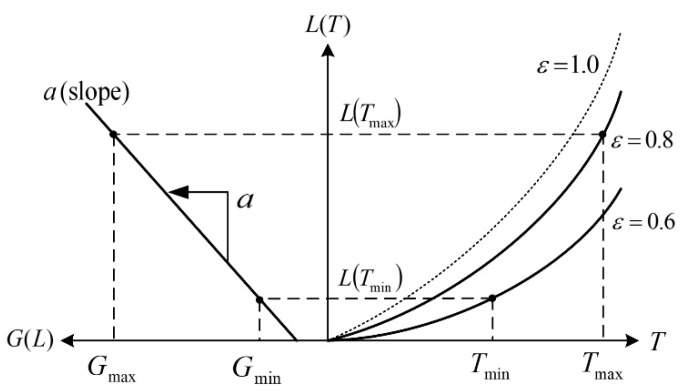
Relation between radiance and gray level of infrared (IR) image according to object temperature.

**Figure 5 sensors-19-02455-f005:**
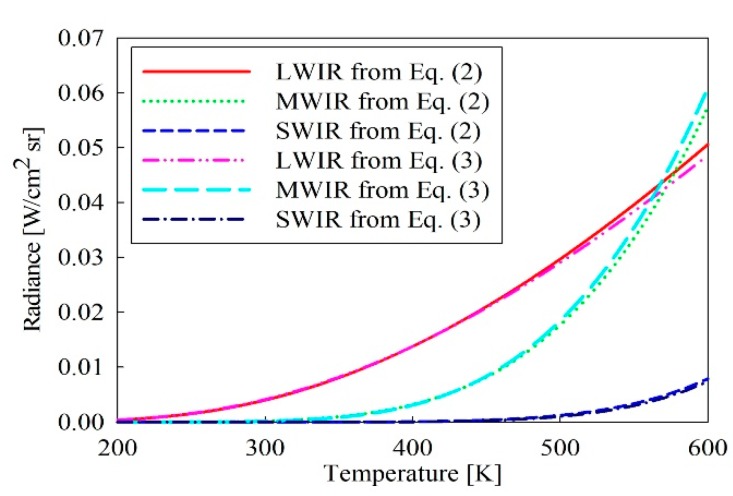
Exact and approximated radiance according to temperature for three IR wavelength bands.

**Figure 6 sensors-19-02455-f006:**
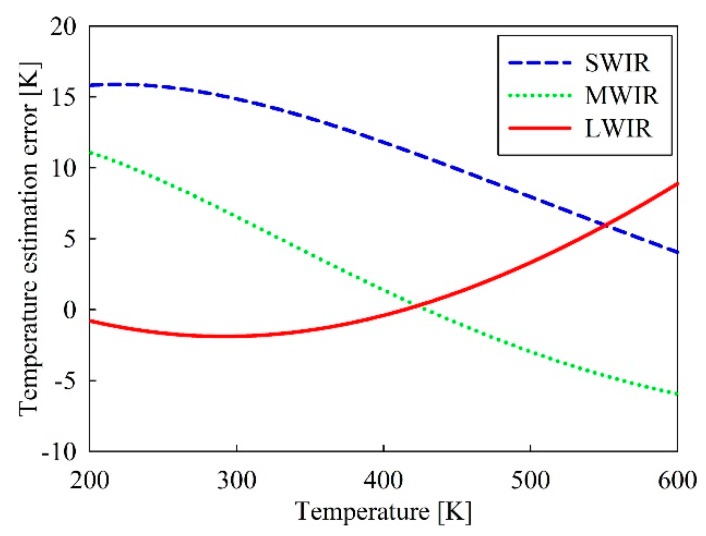
Error of estimated temperature according to temperature for three IR wavelength bands.

**Figure 7 sensors-19-02455-f007:**
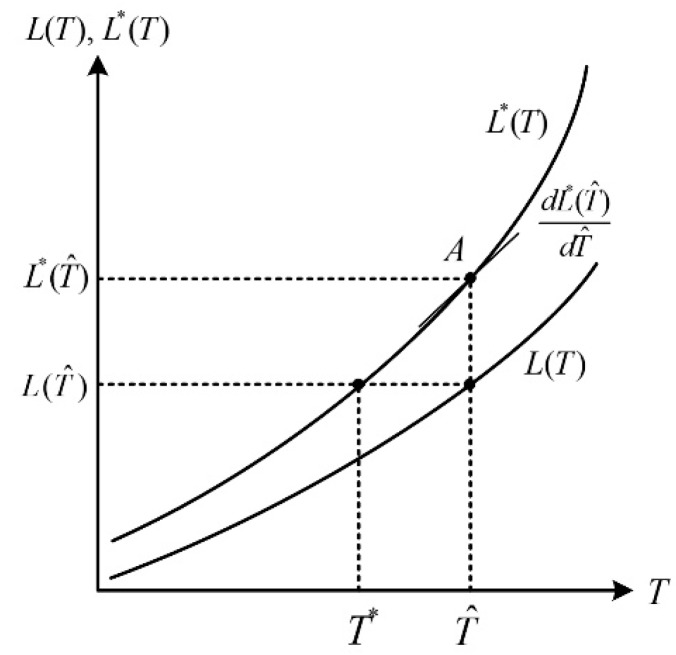
Exact and approximation radiance according to temperature.

**Figure 8 sensors-19-02455-f008:**
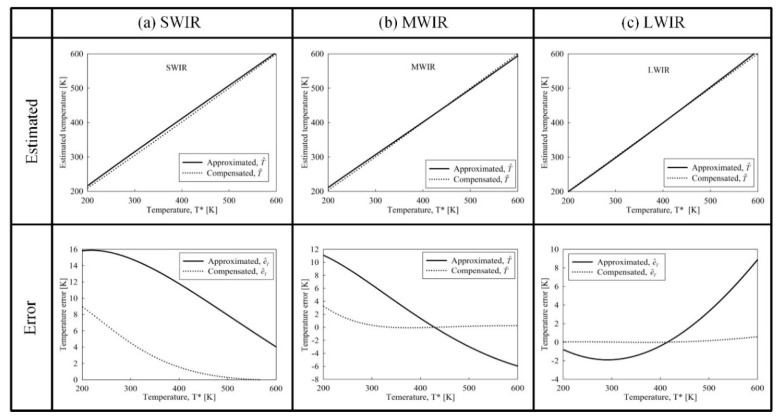
Approximately estimated and compensated temperature and temperature error for (**a**) short wavelength IR (SWIR), (**b**) MWIR-, and (**c**) LWIR-band of 200 K to 600 K.

**Figure 9 sensors-19-02455-f009:**
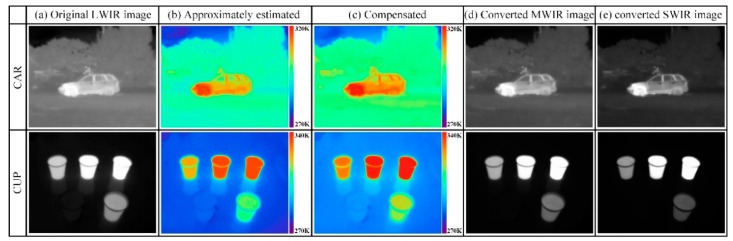
(**a**) Original LWIR image, (**b**) approximately estimated temperature, pseudo color image of (**c**) compensated temperature, (**d**) converted MWIR image, and (**e**) converted SWIR image for CAR and CUP.

**Figure 10 sensors-19-02455-f010:**
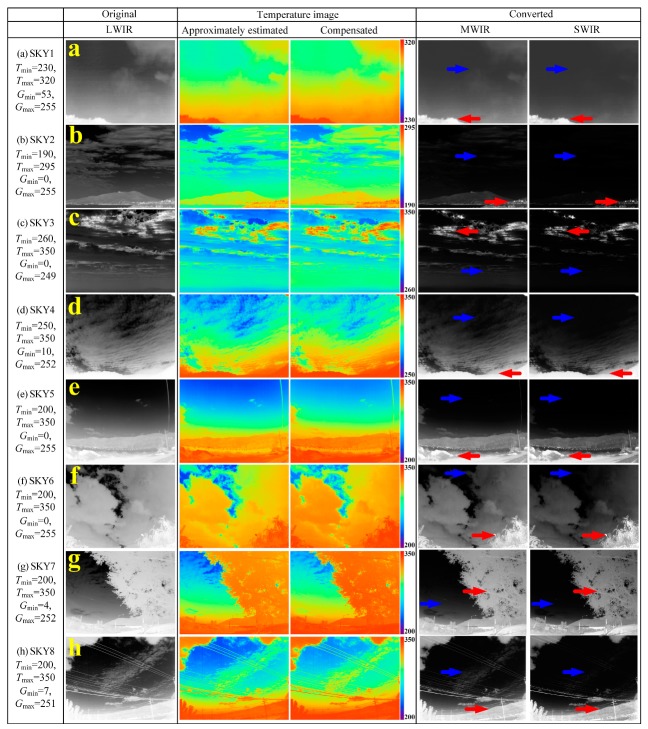
MWIR and SWIR conversion results using the proposed method for various sky background images, (**a**–**h**) SKY1~SKY8.

**Figure 11 sensors-19-02455-f011:**
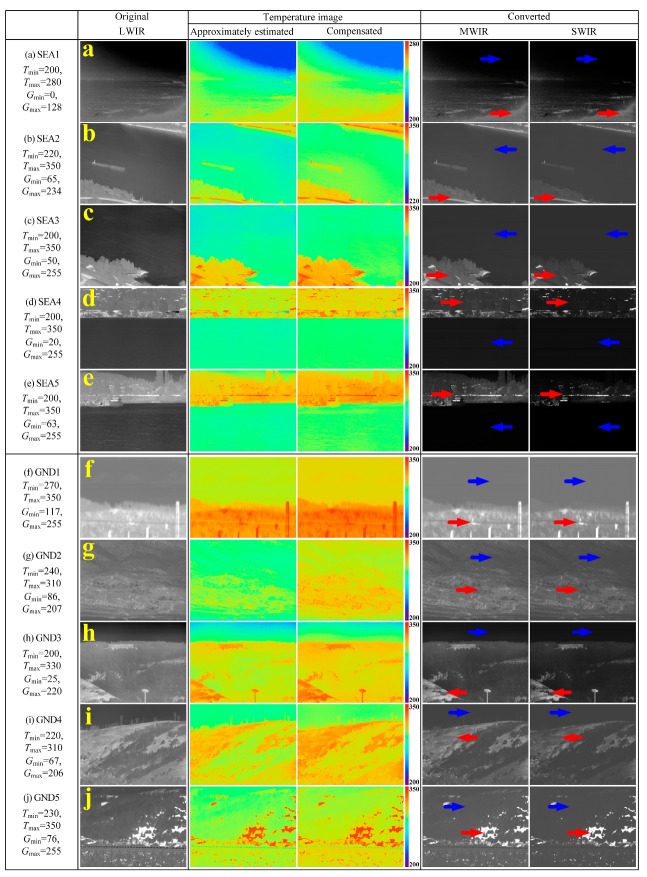
MWIR and SWIR conversion results using the proposed method of various sea and ground background images, (**a**–**e**) SEA1~SEA5 and (**f**–**j**) GND1~GND5.

**Figure 12 sensors-19-02455-f012:**
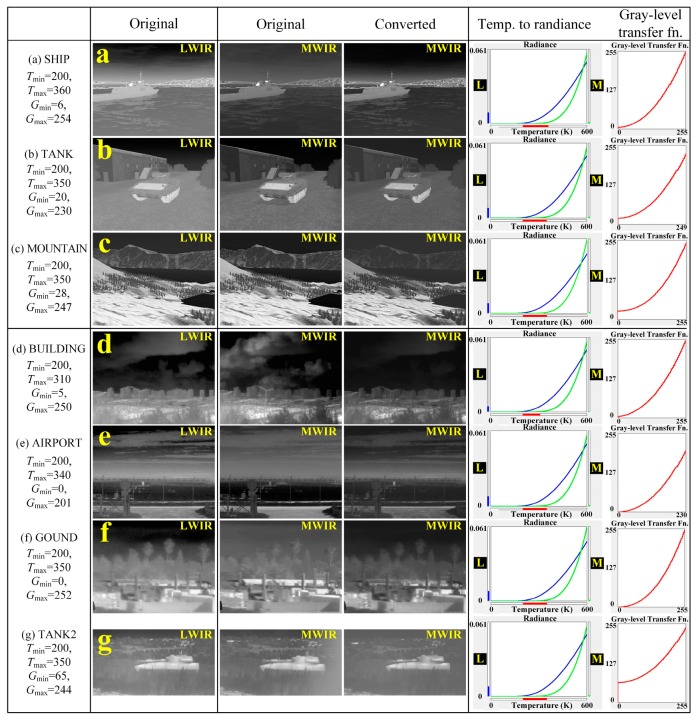
Comparison of converted MWIR image by proposed method and modeled MWIR images ((**a**) SHIP, (**b**) TANK, and (**c**) MOUNTAIN) made by IR sensor simulator and actual MWIR images ((**d**) BUILDING, (**e**) AIRPORT, (**f**) GROUND, and (**g**) TANK2).

**Figure 13 sensors-19-02455-f013:**
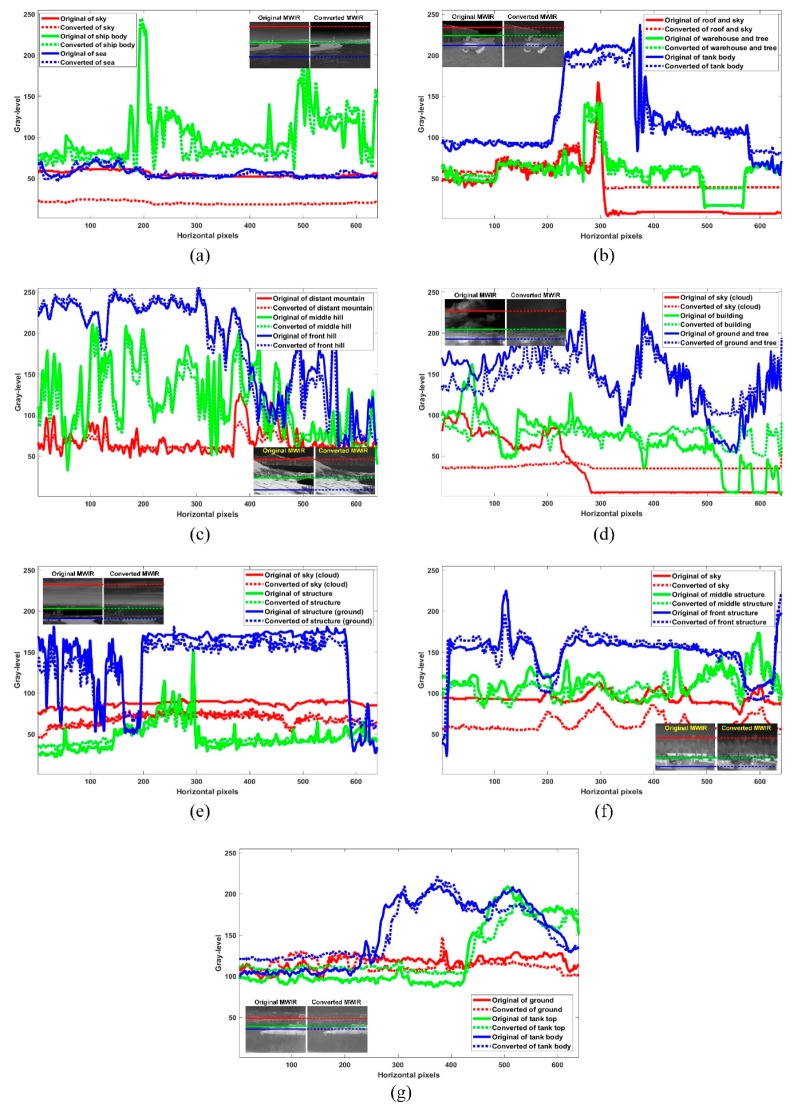
Pixel values of red, green, and blue lines on original MWIR images and converted MWIR image for test images, (**a**) SHIP, (**b**) TANK, (**c**) MOUNTAIN, (**d**) BUILDING, (**e**) AIRPORT, (**f**) GROUND, and (**g**) TANK2.

**Table 1 sensors-19-02455-t001:** PSNR result of red, green, and blue line on original MWIR images and converted MWIR image of Figure 13.

Test Images	Red Line	Green Line	Blue Line
SHIP	17.20 dB	27.93 dB	36.51 dB
TANK	20.92 dB	29.89 dB	29.02 dB
MOUNTAIN	28.45 dB	26.10 dB	27.67 dB
BUILDING	16.68 dB	25.55 dB	25.91 dB
AIRPORT	22.45 dB	31.62 dB	27.00 dB
GROUND	18.38 dB	28.36 dB	28.50 dB
TANK2	28.55 dB	27.65 dB	28.27 dB

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
