# Peer review of "IR-Band Conversion of Target and Background Using Surface Temperature Estimation and Error Compensation for Military IR Sensor Simulation"

_sensors, 2019, doi:10.3390/s19112455_

Round 1

Reviewer 1 Report

The surface temperature of the object is estimated by an approximated Planck’s radiation equation and the error of estimated  temperature is corrected by using a slope information of exact radiance and approximated one. This work is very important and for generating another IR wavelength-band image.

The principle is clear , Sufficient experimental data are provided and the results evaluation is reasonable in this paper.

Author Response

Point 1: The surface temperature of the object is estimated by an approximated Planck’s radiation equation and the error of estimated temperature is corrected by using a slope information of exact radiance and approximated one. This work is very important and for generating another IR wavelength-band image.

The principle is clear, Sufficient experimental data are provided and the results evaluation is reasonable in this paper. 

Response 1: Thank you for your valuable review. Based on your advice, the English grammar and spelling of the paper was checked by native speakers. Please refer to the revised file. Thank you.

Reviewer 2 Report

Manuscript Overview

This manuscript proposed an Infrared (IR) wavelength-band conversion method, which transforms arbitrary IR wavelength band image to another IR wavelength-band based on the surface temperature estimation of an object and the error attenuation method for the estimated temperature. The surface temperature of the object is estimated by an approximated Planck’s radiation equation and the error of the estimated temperature is corrected by using a slope information of exact radiance and its approximation. The corrected surface temperature is used for generating another IR wavelength-band image. The verification of the proposed method is demonstrated through the simulations using actual IR images obtained by thermal equipment.

Few comments needs to be incorporated

Its better to represent the methodology using a flow chart to give reader a quick understanding of the proposed idea.

Its better to represent the dataset used before the results and discussion section in order to avoid confusion.

Clearly mention the significance of a study and highlight it in the conclusion section.

Author Response

Point 1: Its better to represent the methodology using a flow chart to give reader a quick understanding of the proposed idea. 

Response 1: We added the following flow chart and simple explanation in section 2.2 before the detail description.

-------------------------------------------------------------------------------------------------------------

Figure 3 shows the flow chart of the proposed wavelength band conversion method. The proposed method first assigns min, max temperature and radiance to the min, max gray-level pixels in an arbitrary input IR wavelength band image, and then estimates the radiance and temperature for gray-levels of all remaining pixels. Using the estimated temperature, the radiance of desired IR band is obtained and then converted to a gray-level.

Point 2: Its better to represent the dataset used before the results and discussion section in order to avoid confusion.

Response 2: We added section 2.3 for explaining dataset used for experiment as the following.

-------------------------------------------------------------------------------------------------------------

2.3. Dataset used for experiment

The IR images used in the simulation were obtained with TAU640, a LWIR camera of FLIR corp. First, photographed CAR (270 K~322 K) and CUP (271 K~340 K) images containing high temperature objects are used to prove the compensation effect of the approximately estimated temperature. Actual photograph of eight sky background LWIR images (SKY1~SKY8), five sea (or river) background LWIR images (SEA1~SEA5), and five ground background LWIR images (GND1~GND5) with various temperature and gray-level distributions are converted to MWIR and SWIR bands and analyzed. LWIR images (SHIP, TANK, and MOUNTAIN) modeled by IR signature and background simulation tools and actual images taken with LWIR (BUILDING, AIRPORT, GROUND, and TANK2) and are converted into MWIR images by the proposed method. Then, the converted MWIR images are compared with the originally modeled MWIR images and the actual photograph MWIR images through gray-level difference and PSNR.

Point 3: Clearly mention the significance of a study and highlight it in the conclusion section.

Response 3: We rewrote the conclusion based on your advice as the following.

-------------------------------------------------------------------------------------------------------------

Modern military IR imaging systems play a vital role in detecting missiles or other threatening military objects. A variety of IR signature simulation tools help designing real IR imaging systems by analyzing the IR radiation of these military objects in multiple IR bands. In these military IR imaging systems and simulation tools, LWIR and MWIR images are the most widely used wavelength bands for analyzing military objects. The studies for conversion between these bands remain at its basic stage. Acquiring IR images of multiple bands require a real complex IR camera system. Also, many IR signature and background simulation tools require numerous parameter settings for multiple wavelength bands. Our presented work on IR wavelength band conversion study can play an important role in overcoming the difficulty of acquiring IR images of various wavelength bands obtained manually. In addition, this method will be helpful for detecting and tracking military objects by using more IR radiation information extracted from original image and converted image in IRST.

Reviewer 3 Report

This paper proposes and describes an IR wavelength-band conversion method which transforms arbitrary IR wavelength-band image to other IR wavelength-band one based on the surface temperature estimation process.

I found the paper to be interesting, but the contribution is not made clear enough in the Introduction part. It is not sufficiently detailed. The novelty, merit and/or contribution of this paper should be clearly shown in Introduction part.

Generally speaking, the paper is well organized and technically correct. The paper appears well-planned, well-structured and consistent with the initial proposal.

Author Response

Point 1: This paper proposes and describes an IR wavelength-band conversion method which transforms arbitrary IR wavelength-band image to other IR wavelength-band one based on the surface temperature estimation process.

I found the paper to be interesting, but the contribution is not made clear enough in the Introduction part. It is not sufficiently detailed. The novelty, merit and/or contribution of this paper should be clearly shown in Introduction part.

Generally speaking, the paper is well organized and technically correct. The paper appears well-planned, well-structured and consistent with the initial proposal. 

Response 1: We added the contribution contents and references in Introduction as the following.

-------------------------------------------------------------------------------------------------------------

The proposed infrared band conversion method may simplify the construction of IR system that consist of various IR band cameras [32], [33] such as forward looking infrared (FLIR) system for obtaining LWIR and MWIR image used for target tracking and detection. This method can be applied to the above-mentioned small target detection field [26]-[31] by using temperature, radiation, and gray-level information obtained from the plurality of IR band images converted from arbitrary IR band images and can be utilized for accurate classification of military targets such as tanks, planes, and warships [34]-[36]. In addition, in IR and visible image synthesis field [37] for object detection, the detection performance can be further improved by combining converted IR band images.

Round 2

Reviewer 2 Report

The comments has been addressed. However, conclusion needs to be revised.

Author Response

Point 1: The comments has been addressed. However, conclusion needs to be revised. 

Response 1: We rewrote the conclusion based on your advice as the following.

-------------------------------------------------------------------------------------------------------------

This paper proposes an IR wavelength-band conversion method which transforms arbitrary IR wavelength-band image to another IR wavelength-band image based on the surface temperature estimation of an object and the error attenuation technique of the estimated temperature. This IR band conversion method can solve the difficulty of building an expensive and complex IR camera system required to obtain IR images of various IR bands in the FLIR or IRST system. Since this method can create IR images of other bands only with IR images of a specific band obtained from a single IR camera, it can be applied to industrial and military fields to analyze IR radiation information emitted by various objects in multiple bands. In particular, the radiation and gray-level information of objects in various IR bands become an important key to detecting and tracking military threats. And IR images of various IR bands obtained by this method can also be utilized in the recently studied IR and visible image fusion field for high image reliability and understandability.
